# Non-Invasive Estimation of Arterial Stiffness Using Photoplethysmography Sensors: An In Vitro Approach [note 1]

**DOI:** 10.3390/s25113301

**Published:** 2025-05-24

**Authors:** Gianluca Diana, Francesco Scardulla, Silvia Puleo, Salvatore Pasta, Leonardo D’Acquisto

**Affiliations:** Department of Engineering, University of Palermo, Viale delle Scienze, Ed. 8, 90128 Palermo, Italy; francesco.scardulla@unipa.it (F.S.); silvia.puleo01@unipa.it (S.P.); salvatore.pasta@unipa.it (S.P.); leonardo.dacquisto@unipa.it (L.D.)

**Keywords:** photoplethysmography (PPG), optical sensors, arterial stiffness, Young’s modulus, pulse wave velocity, experimental setup

## Abstract

With advancing age, blood vessels undergo deterioration that causes structural and functional changes, including a progressive increase in arterial wall stiffness. Since arterial stiffness is closely linked to the potential risks of cardiovascular diseases, which remains the leading cause of global mortality, it has become essential to develop effective techniques for early diagnosis and continuous monitoring over time. Photoplethysmography, a low-cost and non-invasive technology that measures blood volume changes, has gained increasing popularity in recent years and has proven to be a potential valuable tool for estimating arterial stiffness. This study employs an in vitro experimental setup designed to simulate the cardiovascular system performing under controlled velocity and pressure conditions, in which silicone phantom models with different geometric and mechanical properties were implemented to evaluate their stiffness using a pair of photoplethysmographic sensors. These were employed to measure the pulse wave velocity, currently considered the reference technique for estimating arterial stiffness, correlated through the well-known Moens–Korteweg equation. Photoplethysmographic sensors were placed at three specific distances to determine an optimal configuration for assessing arterial stiffness. Results showed the best performance for softer vascular models at a 15 cm sensor distance, with measurements demonstrating satisfactory accuracy. Variability and standard deviation values increased with model stiffness. The aim of this study is to improve the use of photoplethysmographic sensors for monitoring the mechanical properties of blood vessels and, therefore, to prevent potential cardiovascular diseases.

## 1. Introduction

Cardiovascular diseases (CVDs) are the leading cause of death worldwide, as reported by the World Health Organization (WHO) within the 2021 World Health Report [1], with an estimated 17.9 million deaths per year, affecting both developed and developing countries [2]. Currently, CVD diagnosis and risk stratification mainly focus on identifying high-risk individuals through common risk factors such as age, blood pressure, and lifestyle (e.g., smoking and obesity) [3]. When subjects identified as potential high-risk show symptoms such as chest pain, shortness of breath, and neurological deficits, they are diagnosed through methods such as ultrasound (cardiac ultrasound and/or ultrasound examination of the superficial vessels in the neck region) [4], computed tomography angiography (CTA) of the head and neck and/or heart region [5,6], or magnetic resonance imaging (MRI) [7]. Unfortunately, none of the techniques mentioned are able to optimally diagnose a substantial portion of the population, especially those at low or moderate risk who may have silent risk factors. Therefore, in recent years, there has been an increasing focus on developing new methodologies both to facilitate the early identification of individuals with cardiovascular risk and to ensure adequate monitoring of any cardiovascular disease already diagnosed.

Several studies have shown that arterial stiffness can be defined as one of the most important risk factors for cardiovascular disease [8]. Arterial deterioration represents a pathophysiological process related to aging, in which the arteries undergo a process of dilation and stiffening, with important hemodynamic consequences [9]. These age-induced changes are significantly accelerated in the presence of additional CVDs and are themselves risk factors for both the onset and progression of cardiovascular diseases [10]. Indeed, arterial stiffness can be related to hypertension [11,12], atherosclerosis [13], and other cardiovascular conditions and, as recent studies have shown [14,15,16], an increase in arterial stiffness can be considered a predictor of major cardiovascular events such as heart attacks and strokes. Therefore, the accurate and continuous measurement of arterial stiffness is essential and should become part of the clinical routine.

The main clinical method of estimating arterial stiffness is based on the measurement of pulse wave velocity (PWV), which is the speed with which the pressure wave propagates along two distinct areas of the body [17,18], and the carotid–femoral segment is considered the “gold standard” measurement site for estimating arterial stiffness [19]. PWV can be measured by the means of space–time measurements, therefore considering the ratio between the length of the segment taken into consideration and the pulse transit time (PTT), which is the time from the initial point to the final point [17]. PWV values progressively increase from the ascending aorta to the abdominal aorta, up to the iliac and femoral arteries, as well as in peripheral areas [20]. This increase in PWV is due to the reduction in elastin content in areas further from the heart, which makes the vessels more rigid [21].

The relationship between PWV and arterial stiffness is described by the Moens–Korteweg equation, theorized by the two Dutch scientists in 1878 and which is still widely accepted by the scientific community. This equation correlates PWV to mechanical and geometrical parameters of the blood vessel [10,22].

The mechanical parameters of blood vessels depend on several factors, including the percentage of elastic and collagen within the vascular walls. This composition varies not only from the central to the peripheral areas of the vessels, but also in relation to the age, health status, and lifestyle of the subject [21,23]. With age, the percentage of elastin, the protein that gives elasticity to blood vessels, tends to decrease. Conversely, an increase in collagen is observed, which is a structural protein that causes greater rigidity of the blood vessel. This vascular aging process can however be more accentuated in the presence of cardiovascular diseases [24,25,26]. Thus, knowledge of the elastic modulus is crucial to understanding the mechanical behavior of blood vessels. In healthy conditions, regions with a greater presence of elastin, such as the aorta, show values between approximately 0.2 and 2 MPa [27,28]. For arteries and veins, reference values are between 0.6 and 3.5 MPa [29,30,31]. In pathological conditions, an increase in the elastic modulus of up to 60% can be observed [32,33]. In the literature, several studies and techniques that aim to estimate arterial stiffness have been proposed. Initially, methods for the assessment of PTT both in vitro [34,35] and in vivo [36,37,38] were analyzed, together with studies on the relationship between PWV and arterial stiffness [39,40] and the use of arterial simulators to simulate different physiological and pathological conditions [41,42,43,44]. Different sensors have been employed for these purposes, such as ultrasound sensors [45,46] and Linear Variable Differential Transformer (LVDT) sensors [35,47], which have allowed researchers to obtain accurate measurements in different experimental and clinical contexts and to improve the understanding of mechanical changes associated with aging and cardiovascular diseases.

In clinical practice, the non-invasive measurement of PWV and, consequently, arterial stiffness is commonly performed by image-based screening tools, such as MRI [48,49] or ultrasound-based systems [17,50]. For the same purpose, there are also some commercial devices capable of detecting different hemodynamic parameters, such as Complior [51], SphygmoCor [52], and PulsePen [53]. Although these technologies are frequently adopted in clinical practice [54], they can be cumbersome, expensive, and operator-dependent, and are not the ideal solution when rapid and early screening is needed. Therefore, interest in detection approaches and new technologies [55] that, in addition to being non-invasive, are easily accessible and usable, has grown significantly in recent years. In this context, current research has focused on the use of photoplethysmographic (PPG) sensors to estimate arterial stiffness.

Photoplethysmography is a non-invasive optical technique that measures changes in blood volume in the arteries and capillaries [56]. Since PPG sensors mainly consist of a LED and a photodetector (PD), they have extremely small dimensions and low costs and have therefore been easily integrated into wearable commercial products such as smartwatches and smartbands [57]. These sensors provide a wide range of physiological parameters such as heart rate and blood oxygenation [58,59] but still have unexpressed potential in measuring further parameters. Ongoing research aims to uncover the new applications and capabilities of PPG sensors, expanding their utility beyond conventional measurements. For instance, Puleo et al. [60] assessed aortic valve function using analytical considerations and PPG sensors positioned upstream and downstream of the valve to estimate the pressure gradient across the valve. In particular, the authors used PPG sensors to better understand the variations in PWV and PTT over the transvalvular pressure gradient range and, therefore, the differentiation between normal and pathological function.

In the last decade, there has been a growing research interest in assessing arterial stiffness using PPG, as highlighted in the comprehensive review by Karimpour et al. [61]. This review outlined recent advances and methods for assessing arterial stiffness, highlighting the potential of PPG as a tool for assessing vascular aging. Ferizoli et al. [62] analyzed the different morphological features of PPG signals, focusing on differences between red and infrared wavelengths, and investigated how these features change with varying vessel stiffness. Several studies using a single PPG sensor mainly focused on deriving the surrogate markers of arterial stiffness, such as the stiffness index (SI), reflection index (RI), or augmentation index (AIx), using the PPG sensors, rather than directly quantifying arterial stiffness using parameters such as Young’s modulus, as performed in this study. For example, Brillante et al. [63] analyzed the SI and RI in healthy individuals aged 18–67 years to investigate the impacts of categories such as age, gender, and race. Similarly, Drapkina et al. [64,65] assessed arterial stiffness in obese patients using the SI, RI, and AIx in order to monitor vascular changes in response to statin therapy. Clarenbach et al. [66] studied the correlation between the SI, derived from photoplethysmography, and the AIx, obtained by arterial tonometry, evaluating the usefulness of these two methodologies in cardiovascular risk stratification.

Ultimately, these sensors, today among the most common in wearable devices, represent a promising compromise, combining the ease of use with the possibility of obtaining repeated and continuous measurements over time.

In this work, we present a study on the estimation of arterial stiffness based on photoplethysmographic sensors. A mock circulation loop able to simulate different cardiovascular conditions was designed for the study scope, and four different silicone phantom models were used to simulate the different mechanical and geometrical properties of blood vessels to derive, by the means of two PPG sensors, the PWV and, consequently, Young’s modulus.

To the best of the authors’ knowledge, there are no studies in the literature that have estimated arterial stiffness in terms of Young’s modulus using two PPG sensors, investigating the optimal configuration for their placement, and employing multiple phantom silicone models with a variable thickness and radius within an in vitro cardiovascular system. Similar research was conducted by Fuiano et al. [47], which, unlike the present study, employed a couple of LVDT sensors in an arterial simulator. Furthermore, their work involved a single natural rubber tube model, where the stiffness, also quantified in terms of Young’s modulus, was adjusted by applying a tensioning system, varying the modulus from 1.39 to 2.47 MPa. Similarly, Njoum et al. [67] examined the capability of PPG sensors to estimate arterial stiffness through a direct quantification of the volume elastic modulus (Ev), a parameter that describes the global mechanical properties of arterial walls. The researchers developed an in vitro cardiovascular system consisting of two arterial models simulating a healthy and a diseased artery model, and the models were tested under different flow conditions, including different stroke volumes and pulse frequencies.

So, given the widespread use of the PPG sensors in wearable devices, enabling the additional measurement of arterial stiffness would lead to significant progress in the diagnostic capacity and prevention of cardiovascular diseases. The aim of this study is to assess the capability of PPG sensors to monitor the mechanical properties of blood vessels through an in vitro simulation.

## 2. Materials and Methods

Arterial stiffness can be estimated from the knowledge of pulse wave velocity [17]. There are two main equations that were used in this study. The first is the well-known, and accepted by the scientific community, Moens–Korteweg equation [22], which correlates the PWV to the mechanical and geometrical parameters of the blood vessel:(1)$$PWV=\sqrt{\frac{Eh}{2r\rho }}$$
where *E* is the circumferential Young’s modulus of the tube wall, *h* is the wall thickness, *ρ* is the liquid density, and *r* is the tube radius.

The second equation, instead, allows the measurements to achieve the PWV through the use of a pair of photoplethysmographic sensors which, placed at a certain known distance Δs, can detect the pulse transit time, which is the time delay Δt of the pulse wave:(2)$$PWV=\frac{\Delta s}{\Delta t}$$

To validate this approach, an experimental setup was implemented, consisting of a mock circulatory loop together with the instrumentation for data acquisition and processing. Four different silicone phantom models were used to simulate the different health conditions of blood vessels. Thus, the models had different mechanical and geometrical properties.

### 2.1. Description of Pulsatile Circuit

The general layout of the experimental setup, depicted in Figure 1, has already been adopted by several research groups [47,68] and was composed of six main components interconnected by silicone tubes and plastic connectors as follows:A pulsatile linear pump (P01-48 × 360F; LinMot, Spreitenbach, Switzerland) for flow generation and to simulate the diastolic and systolic phases of the cardiac cycle;An adjustable compliance chamber, used to replicate arterial compliance;An electromagnetic flowmeter (Optiflux 5300C; Krohne, Duisburg, Germany) for monitoring the flow, which was kept constant at 5 L/min;A commercial pressure transducer (X5072 Druck; GE Measurement & Control, Agrate Brianza, Italy) for monitoring systemic pressure, which was maintained within 70 and 120 mmHg;Each replaceable silicon phantom model (length: 50 cm), on which the PPG sensors were positioned for data acquisition and validation of the proposed methodology;A fluid collector containing approximately 3 L of distilled water.

The peripheral resistance and systemic pressure were varied using an adjustable valve placed at the outlet of the silicon model.

Through the LinMot Talk software (v.6.7, LinMot, Spreitenbach, Switzerland), it was possible to set the stroke of the pump (linear position of the piston) as well as the frequency, which allowed us to simulate a specific physiological condition in terms of the stroke volume and heart rate. Specifically, the pump parameters (Figure 2) were configured to obtain a frequency of approximately 90 cycles per minute and a cardiac output of 5 L/min. Then, the additional components (2, 3, 8, and 9 depicted in Figure 1) were calibrated to maintain the pressure along each cycle within 70 and 120 mmHg.

PPG signals were acquired by placing a pair of sensors (DFRobot; Beijing, China), working at a 520 nm wavelength, at three specific distances, 10 cm, 15 cm, and 20 cm, on each of the four silicone models. Each distance (Figure 3) was determined by means of a caliper (Universal vernier models; TESA Technology, Renens, Switzerland) considering the PD–PD distance. These values were selected to ensure sufficient temporal separation of the pulse wave peaks for accurate PTT estimation, while remaining compatible with typical anatomical sites, such as the forearm, where such sensor spacing is physically applicable. During the experiments, the PPG sensors were tied to the silicone phantom model, ensuring a constant and reliable contact pressure. The setup was designed to minimize external interference and maintain a stable baseline for the measurements.

### 2.2. Data Acquisition and Processing

Signals were acquired using a data acquisition board (NI USB-6009, National Instruments, Austin, TX, USA) for real-time acquisition with a sampling frequency of 5 kHz and a dedicated algorithm implemented through LabVIEW software (v.2021, National Instruments, Austin, TX, USA).

Data processing was performed using Matlab software (v.2022b, The MathWorks Inc., Natick, MA, USA). Initially, signal filtering was applied to the raw signals using a fourth-order Butterworth bandpass IIR filter, with frequencies ranging from 0.5 to 5 Hz. These values were selected based on a preliminary frequency–domain analysis of the PPG signals, aimed at isolating the fundamental components of the cardiac waveform while effectively removing low-frequency drift and high-frequency noise. Subsequently, PTT was achieved through two different peak detection approaches that were further compared.

The first methodology used was the peak-to-peak method, which consists in identifying the main peaks of the two curves using the findpeaks algorithm (Figure 4A). The second methodology used was the tangent method, where the PTT is estimated by calculating the time distance between two points of the maximum slope of the ascending phase of the signal. In this case, the findpeaks algorithm was applied to the first derivative of the signal (Figure 4B).

Once the PTT was estimated, the value of the PWV was calculated using Equation (2), as the distance of the sensors positioned on the phantom model was known. Subsequently, the experimental value of Young’s modulus was calculated using the inverse Equation (1).

### 2.3. Experimental Protocol

Each silicone phantom model was placed within the circuit and then the two PPG sensors were positioned at one of the three predetermined distances on the model. This placement was a crucial part of the study for the accuracy of the PPT, from which the PWV and, consequently, Young’s modulus were derived.

Then, the pump was activated, any air bubbles were removed, and it was ensured that the system operated within the predetermined physiological range by monitoring the flow rate (5 L/min) and the pressure values detected by the sensor, ranging from 70 mmHg (diastolic) to 120 mmHg (systolic).

Signals were acquired using a data acquisition board and a dedicated algorithm on LabVIEW (2.2) (v.2021, National Instruments, Austin, TX, USA). The experiments were repeated several times under identical conditions to ensure the reliability and reproducibility of the results. Ten measurements lasting 3 min for each experiment were acquired, for a total of 120 tests.

### 2.4. Uniaxial Tensile Test

To compare the experimental values obtained with the reference ones, uniaxial tensile tests were performed on a total of 12 specimens for each of the 4 models proposed in the study. These samples were firmly clamped between the grips of a tensile testing machine (Figure 5A), ensuring proper alignment and uniform stress distribution during the test.

To accurately measure the deformation and calculate Young’s modulus, two black markers were placed on each specimen at one-quarter and three-quarters of the sample’s length using a caliper (Figure 5B). These markers served as the reference points for tracking the displacement during the test (Figure 5C). A high-definition camera (C910, Logitech, Switzerland) was positioned orthogonally to track, in real time, the displacement of the markers during the stretch and, thus, the material’s deformation. In particular, a block diagram was created using the LabVIEW Vision Development Module, as described by Pasta et al. [69]. This allowed us to track the centroid of the markers by pixel-weighted operation through pattern matching of the images. Then, the images, collected at 30 frames per second, were converted to a monochromatic scale and scaled to 380 × 240 pixel resolution. Each specimen was subjected to a controlled elongation of 20 mm, with a machine crosshead speed of 2 mm/min. The output data comprised the load-cell force signals and the X–Y coordinates of each marker. Data were analyzed through Matlab to obtain the unidirectional stress and strain calculation. Strain, both in the x- and y-directions, was computed based on the relative displacement of the markers from their initial positions. For each axis, the mean displacement was calculated by averaging the distance between the paired markers. Strain values were then determined as the normalized displacement relative to the initial marker positions. Stress was calculated by dividing the recorded force values by the cross-sectional area of the sample, derived from the input thickness and width. Stress–strain curves were then generated to analyze the material’s response under loading. To determine Young’s modulus for each test, the slope of the linear portion of the stress–strain curve was calculated, representing the elastic region of the material. This slope was obtained by fitting a linear regression to the initial linear segment of the stress–strain data. The Young’s modulus values obtained from each test were averaged to provide a representative modulus for each silicon model.

## 3. Results

The aim of this study was to investigate the relationship between arterial stiffness, quantified by Young’s modulus, and the accuracy of the pulse transit time at different fixed distances between the sensors. Initially, for all tests, the PTT was obtained from the temporal difference between the peaks detected with the findpeaks algorithm on the original PPG waveforms. However, it was observed that with a distance of 10 cm between the sensors, this approach did not allow the measurements to achieve reliable values, with an average error of approximately 20%, excluding Model 4, which showed the worst performance using the peak-to-peak method at 10 cm, consistently producing underestimated and incoherent values that significantly deviated from the reference ones. Thus, for the shorter distance (i.e., 10 cm), the tangent method was adopted, which proved to be more reliable for the measurements. In order to compare the two measurement methodologies, the tangent method was also applied for the distances of 15 cm and 20 cm.

The results of Young’s modulus, obtained both through the uniaxial tensile test and using the PPG sensors with the two methodologies, are reported, respectively, in Table 1 and Table 2.

It was possible to observe that the standard deviation increased as the model became more rigid. This trend was better visualized in Figure 6 and Figure 7, which shows the relationship between Young’s modulus and the increase in the stiffness of the silicone models with error bars.

Therefore, to better evaluate the performance of each method under different conditions, the most suitable combination of methodology and distance was considered for each case. Based on the results obtained, the tangent method proved to be more robust at the shortest distance (10 cm), likely due to its lower sensitivity to wave reflections and noise. Conversely, at greater distances (15 cm and 20 cm), the peak-to-peak method provided more accurate and consistent estimates of Young’s modulus. As a result, in the following comparative analyses, results were selected based on the best-performing method at each distance: tangent–secant at 10 cm and peak-to-peak at 15 and 20 cm.

The Bland–Altman analysis (Figure 8) visually confirmed the results, showing a narrower dispersion in the softer models (Models 1 and 2), while there was a greater dispersion in the stiffer models (Models 3 and 4) and, hence, an increase in the variability in the results as the model stiffness increased.

Figure 9 shows the box plots of the Young’s modulus values calculated for the 12 experimental cases (the four models at the three fixed distances). In each graph, the values for the ten experimental tests are represented, where each box plot was built starting from the values obtained in the time interval of the single test. Specifically, the Young’s modulus values were considered to be calculated every 5 s (each test lasts 3 min) with the Moens–Korteweg equation.

The red dotted line within each graph represents the reference value obtained from the tensile tests for each model. This comparison showed how the values measured during the tests were distributed with respect to the reference values, highlighting the variation and precision of the measurements at different distances.

To establish the agreement between the experimental methodology based on PPG and the reference values (i.e., the tensile test), the standard error (SE) was calculated for all cases, i.e., for the four models at the three distances. As reported in Table 3, it was observed that the highest uncertainty values were recorded for the stiffest models.

To compare the experimental methodology of this study, based on the PPG sensors, with the reference methodology, the Pearson correlation coefficient was measured, as shown in Table 4. The mean values obtained in each test and the values obtained from the tensile tests were considered and the ratio between the covariance of the two variables and the product of their standard deviations was analyzed.

To contextualize the experimental results of the Young’s modulus values, Figure 10 presents the pressure waveforms acquired by the pressure transducer for the four models used in the study. The pressure values were recorded during the acquisition of the PPG signals and remained constant for each model in all the tests performed. The different shapes depend on the radius and on the elasticity of each model, which caused a slight variation of the pressure wave within the circuit.

## 4. Discussion

In this study, experimental tests were performed, reproducing an in vitro cardiovascular system. A pair of PPG sensors were used on the surface of four silicone phantom models with differential mechanical and geometric properties to estimate their mechanical parameters and, ultimately, evaluate their stiffness. The parameter that significantly affects the estimation of arterial stiffness is the PWV or, equivalently, the PTT [17]. In particular, an increase in the mean PWV (or a decrease in the PTT) is associated with an increase in arterial stiffness [70]. This relationship is explained by the Moens–Korteweg equation (Equation (1)) which shows how the PWV is proportional to the square root of Young’s modulus, the reference quantity to quantify arterial stiffness [10]. This approach, previously introduced by the authors in earlier work [71], represents a preliminary contribution on the potential effectiveness of PPG sensors in estimating Young’s modulus, which may be implemented in the future in the design of wearable devices. It should be noted, however, that in an in vivo scenario, several factors, such as vascular tone and endothelial activity, can influence the quality and accuracy of the measurements, which cannot be replicated in an in vitro setting.

Based on the approach adopted in this study, the authors observed that the reliability in PTT estimation (Equation (2)) was strongly dependent on the distance between the PPG sensors. These distances were selected to determine whether an optimal configuration existed for accurately estimating the stiffness of the models. The results showed that while the peak-to-peak method (Figure 4A), initially adopted as the primary approach for estimating PTT, produced consistent and accurate results at the sensor distances of 15 cm and 20 cm, a significant increase in percentage errors was observed at a 10 cm distance across all tests (Table 1). This discrepancy is likely due to the proximity of the sensors and the potential influence of wave reflection phenomena. In particular, the piston pump employed to simulate the flow in the circuit, despite maintaining a continuous cycle, may introduce noise or backward pressure fluctuations due to its valve action. These noises could amplify the effect of the reflected wave, especially at shorter sensor distances, making it difficult to isolate the forward wave from the reflected wave, thus affecting the accuracy of the PTT measurement. Therefore, to address this issue, the tangent method (Figure 4B) was applied to the data acquired at 10 cm. This approach focuses on the ascending slope of the signal and has been shown to be less susceptible to interference from reflected waves. By analyzing the initial rise of the waveform, the tangent method provided a more reliable estimate of the PTT for the 10 cm distance, reducing the impact of wave reflection and proximity effects on the measurements. This methodological adjustment allowed more consistent and accurate PTT estimates at the shorter sensor distance. It is worth noting, however, that while the tangent method offers greater robustness in noisy or reflective environments, it is also computationally more complex, as it requires calculating the derivative of the waveform and identifying the intersection point of the tangent with the time axis. Conversely, the peak-to-peak method, based on identifying the maximum points of the waveform, is significantly simpler to implement and computationally lighter. Interestingly, in this study, the peak-to-peak method yielded slightly more accurate results at longer sensor distances (15 cm and 20 cm), suggesting that under favorable signal conditions, it may be a valid and efficient alternative to more complex approaches. Both methods have been extensively discussed by Filippi et al. [34,72], who also highlighted the robustness of a third approach based on cross-correlation. This technique, which assesses the degree of similarity between two waveforms, is suggested as a more robust method for PTT evaluation and may serve as an inspiration for future implementation.

The most relevant results of this study highlighted an optimal positioning of the PPG sensors at a distance of 15 cm. Furthermore, an increase in the standard deviation in the final calculation of Young’s modulus was observed with increasing stiffness of the models for all four distances and for both methodologies, as shown in Table 1 and Table 2. These results are consistent with those obtained in the study by Fuiano et al. [47], where an increase in the standard deviation corresponded to an increase in the stiffness of the model. This phenomenon was attributed by the authors to the viscoelastic nature of the silicone model used.

To compare the experimental and reference values, tensile tests were conducted on the models to obtain the reference elastic modulus (Table 1 and Table 2). The calculation of the standard error and an analysis of the Pearson correlation coefficients were performed to compare the reference methodology (tensile test) with the experimental methodology based on the use of the PPG sensors. The analysis of the standard error (Table 3) revealed lower uncertainty values for Models 1 and 2 (the softest), while these values were higher for the models with higher stiffness. Similarly, the results aligned with those reported by Fuiano et al. [47] in Table 4 of their study, which demonstrated an increase in percentage error with higher tensioning states.

Significant results were obtained from the analysis of the Pearson correlation coefficients (Table 4), which showed values higher than 0.95 for the softer models (Models 1 and 2), indicating a strong correlation between the results of the two methodologies, while this correlation tended to decrease as the stiffness of the models increased. Similarly, the Bland–Altman plot (Figure 8) showed a more compact and concentrated distribution of the points in the softer models, indicating greater coherence and less variability of the results. In contrast, greater dispersion of the points was observed in the stiffer models (Models 3 and 4), reflecting an increase in measurement variability as the rigidity of the model increased. Another graphic representation, based on the use of box plots (Figure 9), allowed us to compare the variability of the results of each test. This representation showed that at a distance of 15 cm, the variability of the results was lower and the accuracy was higher.

These findings aligned with those reported by Hong et al. [73], who evaluated the ability of 19 indices related to vascular aging stiffness to assess aortic stiffness in a cohort of approximately 4000 healthy subjects aged between 25 and 75 years. While their study quantified arterial stiffness using aortic pulse wave velocity (aoPWV), the results were comparable to those of the present study. Notably, the authors compared six pulse wave velocity indices with the theoretical aortic pulse wave velocity (aoPWVt), and the results showed increased dispersion with advancing age, reflecting the physiological increase in arterial stiffness. Furthermore, their analysis of the Pearson correlation coefficients between each vascular aging index and aortic Young’s modulus demonstrated a marked decline in correlation among the older subjects. This trend was aligned to the findings of the current research.

In conclusion, although an optimal distance of 15 cm has been highlighted, further studies will be performed and compared with in vivo data to conduct experiments with new and more accurate data processing techniques. Based on the results obtained, the proposed methodology could contribute to the early identification of cardiovascular health through an accurate estimation of arterial stiffness.

### Limitations

This study confirms the capability of PPG sensors to provide a direct estimation of arterial stiffness. However, several limitations must be acknowledged before this method can be applied to in vivo investigations, with the primary limitation being the in vitro nature of the study.

Arterial stiffness is influenced by the fluid density and the geometrical parameters of the arterial segment, namely radius and thickness. In a real clinical setting, the availability of these two parameters (radius and thickness) is not always guaranteed. To simulate the different physiopathological and, consequently, biomechanical conditions of blood vessels, this study employed cylindrical silicone models, which do not perfectly replicate the viscoelastic behavior of arterial walls. Additionally, the radius and thickness of the silicone models are larger than the anatomical ones. As for the fluid, distilled water was used, whose physical properties are slightly different from those of human blood, which has a higher density (∼1050–1060 kg/m^3^ for blood vs. ∼1000 kg/m^3^ for distilled water) and viscosity (∼3–4 mPa·s for blood vs. ∼0.7 mPa·s for distilled water at 37 °C) due to the presence of cellular components, such as erythrocytes and plasma proteins, as well as non-Newtonian behavior. As a result, the optical properties of blood differ from those of water, potentially influencing the measurements performed with PPG sensors.

In a potential in vivo scenario, further challenges must be considered. Human skin introduces additional optical attenuation and scattering, which varies with skin tone and subcutaneous fat thickness, potentially affecting signal quality. Moreover, motion artifacts due to patient movement, breathing, or muscle contractions may introduce noise into the PPG signal, complicating the estimation of arterial stiffness.

Despite these limitations, the method proposed in this study was capable of providing an indirect measurement of Young’s modulus, with acceptable accuracy when compared to the gold standard of tensile testing. This approach could represent a first step for the further development of non-invasive detection systems for the continuous monitoring of arterial stiffness and, consequently, cardiovascular health assessment.

## 5. Conclusions

This work focused on the development and testing of an experimental setup based on PPG sensors and a hydraulic circuit to estimate the Young’s modulus values of four silicone models simulating different blood vessel health conditions, providing significant insights into the optimal sensor placement for arterial stiffness monitoring and PTT analysis methodologies. Young’s modulus estimation was based on the Moens–Korteweg equation, widely accepted in both the medical and scientific fields, which correlates Young’s modulus with the pulse wave velocity, which in turn can be estimated from spatiotemporal measurements. The obtained results highlighted that the best performance in estimating Young’s modulus was observed for the softer models at a sensor distance of 15 cm, where the measurements demonstrated proper accuracy, which tended to decrease with increasing stiffness. These results proved to be consistent with those found in the literature, confirming the reliability of the experimental approach and supporting the development of a low-cost monitoring system for the estimation of arterial stiffness and cardiovascular health.

## Figures and Tables

**Figure 1 sensors-25-03301-f001:**
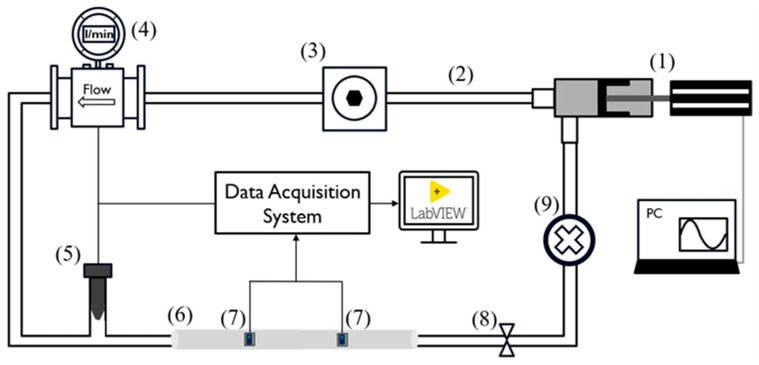
Experimental setup scheme. The numbers reported refer to the following components: pulsatile pump (1), silicone tubes (2), compliance chamber (3), flowmeter (4), pressure transducer (5), silicon phantom model (6), PPG sensors (7), adjustable valve (8), and fluid collector (9).

**Figure 2 sensors-25-03301-f002:**
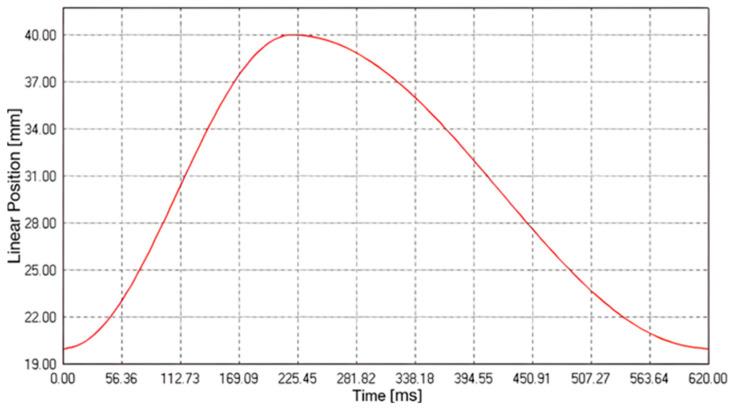
Pump cycle curve to simulate the systolic and diastolic phases of the cardiovascular system.

**Figure 3 sensors-25-03301-f003:**
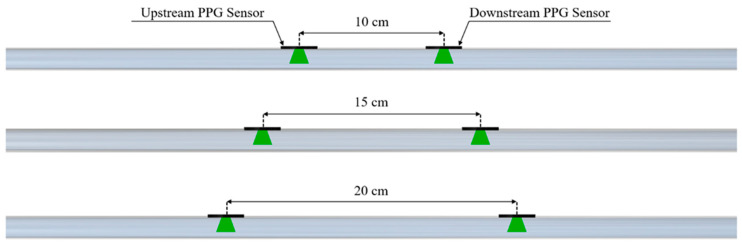
Configurations of the PPG sensors on the phantom model for PTT detection.

**Figure 4 sensors-25-03301-f004:**
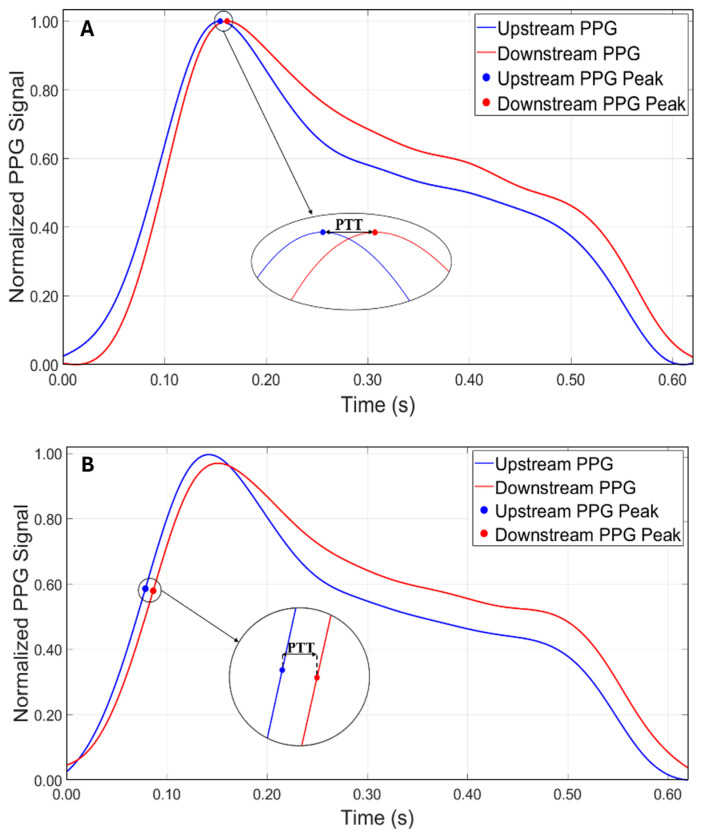
Comparison between the two approaches for calculating PTT. (**A**) Peak-to-peak method. (**B**) Tangent method.

**Figure 5 sensors-25-03301-f005:**
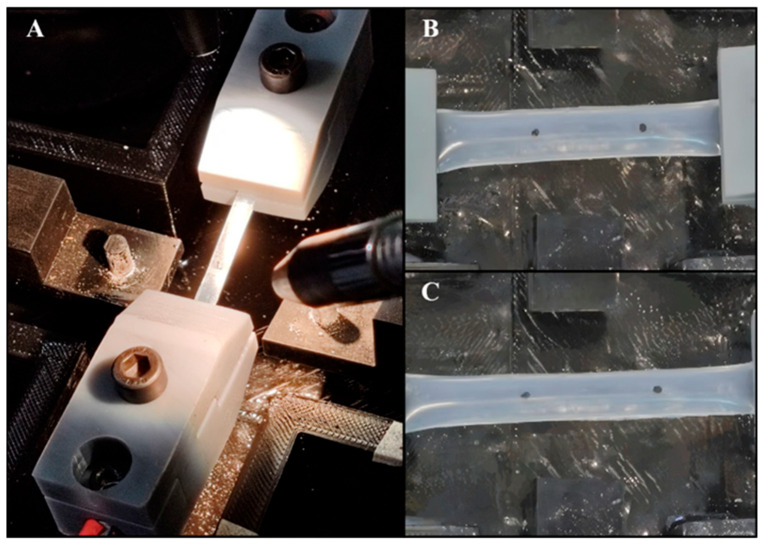
(**A**) Tensile testing machine setup. (**B**) Silicone specimen at rest, showing initial marker positions. (**C**) Silicone specimen post-tensile test, indicating the displacement of the markers and elongation of the specimen.

**Figure 6 sensors-25-03301-f006:**
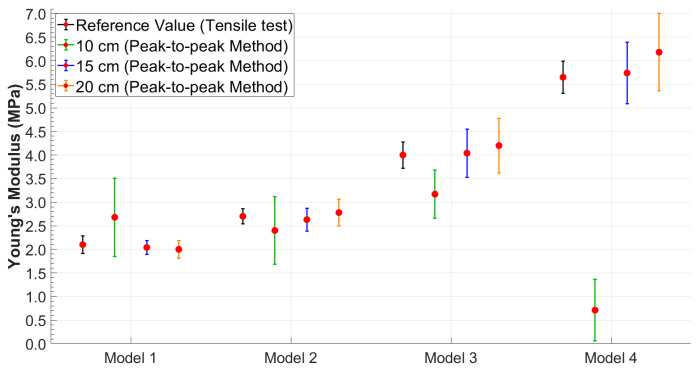
Young’s modulus values obtained using the peak-to-peak method with error bars highlighting data variability.

**Figure 7 sensors-25-03301-f007:**
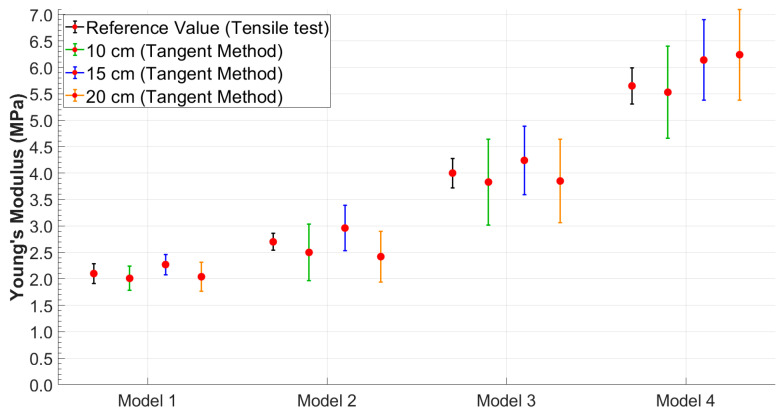
Young’s modulus values obtained using the tangent method with error bars highlighting data variability.

**Figure 8 sensors-25-03301-f008:**
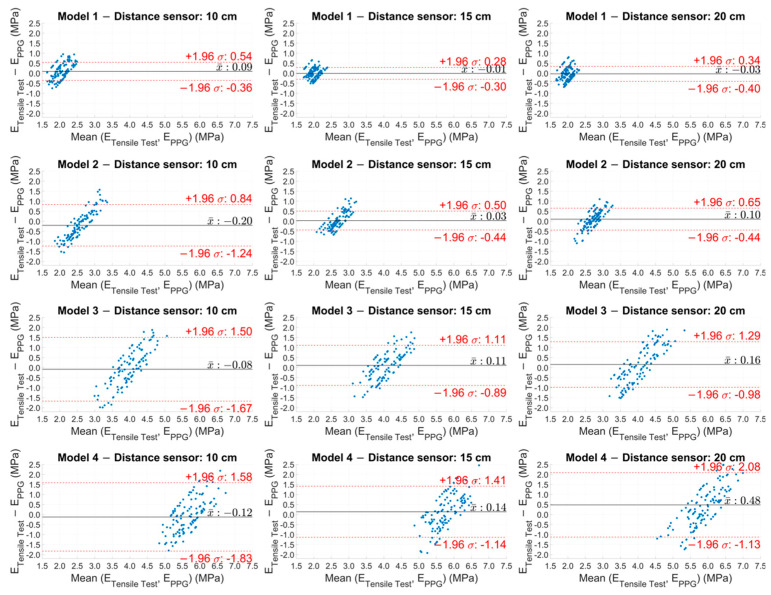
Bland–Altman plot of the experimental Young’s modulus values estimated with PPG sensors for the four models at the three distances.

**Figure 9 sensors-25-03301-f009:**
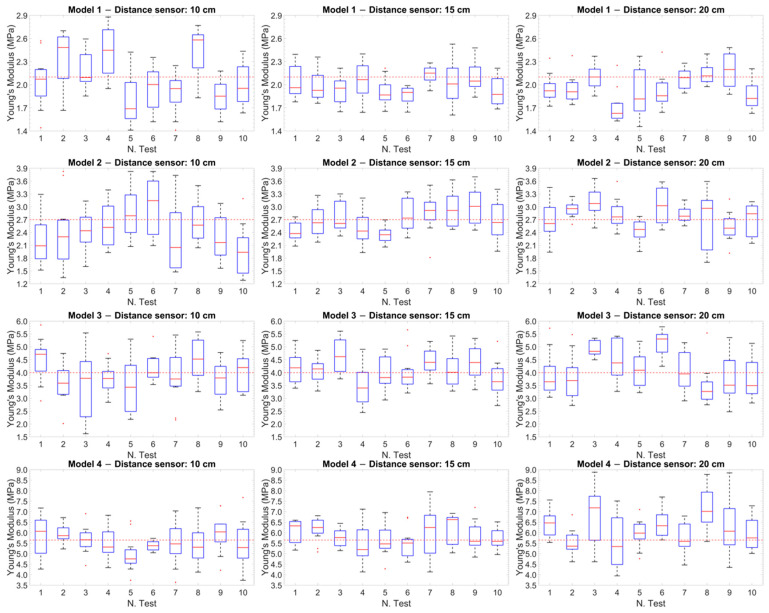
Box plot of the Young’s modulus values calculated for the 12 experimental cases. The red dotted line within each plot represents the reference value obtained from the tensile tests for each model.

**Figure 10 sensors-25-03301-f010:**
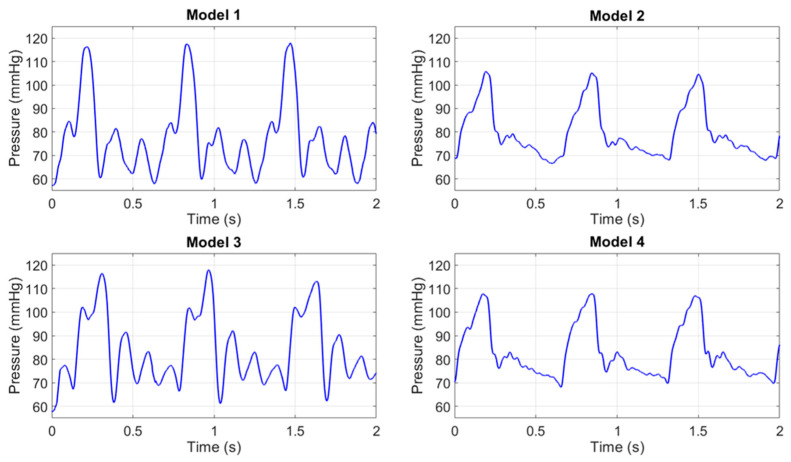
Pressure waves in the 4 models.

**Table 1 sensors-25-03301-t001:** Mechanical and geometrical properties of the models and the experimental Young’s modulus values obtained using the peak-to-peak method at the three distances.

N.	Radius (mm)	Thickness (mm)	Tensile Test (MPa)	Experimental Value 10 cm (MPa)	Experimental Value 15 cm (MPa)	Experimental Value 20 cm (MPa)
1	4.30	1.30	2.10 ± 0.19	2.68 ± 0.83	2.04 ± 0.15	2.00 ± 0.19
2	7.50	1.50	2.70 ± 0.16	2.40 ± 0.72	2.63 ± 0.24	2.78 ± 0.28
3	4.00	1.00	4.00 ± 0.28	3.17 ± 0.51	4.04 ± 0.51	4.20 ± 0.58
4	8.00	2.00	5.65 ± 0.34	0.71 ± 0.65	5.74 ± 0.65	6.18 ± 0.82

**Table 2 sensors-25-03301-t002:** Mechanical and geometrical properties of the models and the experimental Young’s modulus values obtained using the tangent method at the three distances.

N.	Radius (mm)	Thickness (mm)	Tensile Test (MPa)	Experimental Value 10 cm (MPa)	Experimental Value 15 cm (MPa)	Experimental Value 20 cm (MPa)
1	4.30	1.30	2.10 ± 0.19	2.01 ± 0.23	2.27± 0.19	2.04 ± 0.27
2	7.50	1.50	2.70 ± 0.16	2.50 ± 0.53	2.96 ± 0.43	2.42 ± 0.48
3	4.00	1.00	4.00 ± 0.28	3.83 ± 0.81	4.24 ± 0.65	3.85 ± 0.79
4	8.00	2.00	5.65 ± 0.34	5.53 ± 0.87	6.14 ± 0.76	6.24 ± 0.86

**Table 3 sensors-25-03301-t003:** Standard error analysis for Young’s modulus values measured with PPG sensors.

N. Model	10 cm (MPa)	15 cm (MPa)	20 cm (MPa)
1	2.01 ± 0.07	2.04 ± 0.05	2.00 ± 0.06
2	2.50 ± 0.17	2.63 ± 0.08	2.78 ± 0.09
3	3.83 ± 0.26	4.04 ± 0.16	4.20 ± 0.18
4	5.53 ± 0.28	5.74 ± 0.21	6.18 ± 0.26

**Table 4 sensors-25-03301-t004:** Pearson correlation coefficients to compare classical methodology (tensile test) and experimental methodology.

N. Model	10 cm	15 cm	20 cm
1	0.95	0.98	0.97
2	0.95	0.97	0.96
3	0.61	0.62	0.61
4	0.42	0.44	0.43

## Data Availability

The raw data supporting the conclusions of this article will be made available by the authors on request.

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
