# Peer review of "Non-Invasive Estimation of Arterial Stiffness Using Photoplethysmography Sensors: An In Vitro Approachâ€"

_sensors, 2025, doi:10.3390/s25113301_

Round 1
Reviewer 1 Report
Comments and Suggestions for Authors
The work presented in your paper is an interesting proposal, based on previous work and experience in the area, to estimate arterial stiffness by means of a non-invasive technique using PGG sensors. This approach is promising since this kind of sensor is easily available and at a low cost. Globally, the paper is well structured, and the scientific procedure is carried out in a proper manner. The format of the article is right, and the English style is correct.
However, I do not really agree with the way the experimental results were presented in Table 1: even if the results obtained at 10cm are not reliable (and it would be interesting to define what does “not reliable” mean, to quantify it) for “peak-to-peak method”, you should present in the table the results for the three distances, applying the same method, in order to compare the obtained results with the same parameters. So, that I propose to you is:
- To keep Table 1, presenting the results for the three distances, obtained via the “peak-to-peak method”.
- To add a new table, presenting the results for the three distances, obtained applying the “tangent secant method” for 10, 15 and 20cm.
- Regarding the Figure 6, you can present the results obtained with the “peak-to-peak method”.
- To include a new figure (7?), presenting the results obtained with the “tangent secant method”, for the three distances.
In my opinion, this will add additional value to your results, because you will be able to study the influence of the two different methods (“peak-to-peak method” vs. “tangent secant method”) on your results, and even to analyze other crucial aspects like the computational load of each one of the method (to be study in depth in future work, perhaps?).
In addition, I provide some other suggestions to enhance your paper:
- There is a text repetition between lines 204 and 211.
- I think that the text between lines 342 and 385 should be placed in the Section 1 “Introduction”, since it makes part of the state-of-the-art, unless I am mistaken.
- A little table summarizing the main advantages and disadvantages of each method presented between lines 87 and 95 will be appreciated.
- To better illustrate the diagram shown in Figure 1, could you provide a picture of the whole real physical testbed (except if confidential)?
- When you analyze the limitations of your work, could you give some details about the densities of distilled water and blood, and about the hypothetical effect of the skin on the measures obtained by the PGG sensors?
Reviewer 2 Report
Comments and Suggestions for Authors
This study presents an innovative in-vitro approach to estimate arterial stiffness using photoplethysmography (PPG) sensors. By simulating cardiovascular conditions with silicone phantom models of varying stiffness, the authors measured pulse wave velocity (PWV) and derived Young’s modulus via the Moens-Korteweg equation. The research highlights the optimal sensor distance (15 cm) for accurate measurements, particularly in softer models, while noting increased variability with higher stiffness. The methodology is well-designed, combining experimental setups with tensile tests for validation. The work contributes to non-invasive arterial stiffness monitoring, with potential applications in wearable devices. However, limitations include the in-vitro nature and simplified fluid properties compared to blood. Overall, the study offers valuable insights into PPG-based vascular health assessment.
- Suggest the authors enhance the intordution part with other technologys, such as Wearable photonic smart wristband for cardiorespiratory function assessment and biometric identification.
-
Suggest clarify the rationale for selecting 10, 15, and 20 cm as sensor distances in the Methods.
-
Suggest Discuss potential in-vivo challenges (e.g., skin tone, motion artifacts) in the Limitations.
-
Suggest add a brief comparison of tangent vs. peak-to-peak methods in the Results.
-
Suggset specify the sampling rate/filter settings for PPG signal processing in more detail.
Round 2
Reviewer 1 Report
Comments and Suggestions for Authors
Thank you for your corrections and explanations following my comments. The current paper has been substantially improved compared with your previous version, and most of the main issues have been correctly addressed.
I have just only one minor comment concerning the current version: the titles of Table 1 and Table 2 are identical, which is not correct. The title of Table 2 should indicate “Tangent method” instead of “Peak-to-peak method”, unless I am mistaken.
Author Response
Dear Reviewer,
Thank you very much for pointing out this oversight. I have now corrected the error accordingly. We truly appreciate your careful review and all the work you have done.